# The ER folding sensor UGGT1 acts on TAPBPR-chaperoned peptide-free MHC I

Lina Sagert, Christian Winter, Ina Ruppert, Maximilian Zehetmaier, Christoph Thomas, Robert Tampé*

Institute of Biochemistry, Biocenter, Goethe University Frankfurt, Frankfurt am Main, Germany

**Abstract** Adaptive immune responses are triggered by antigenic peptides presented on major histocompatibility complex class I (MHC I) at the surface of pathogen-infected or cancerous cells. Formation of stable peptide-MHC I complexes is facilitated by tapasin and TAPBPR, two related MHC I-specific chaperones that catalyze selective loading of suitable peptides onto MHC I in a process called peptide editing or proofreading. On their journey to the cell surface, MHC I complexes must pass a quality control step performed by UGGT1, which senses the folding status of the transiting N-linked glycoproteins in the endoplasmic reticulum (ER). UGGT1 reglucosylates non-native glycoproteins and thereby allows them to revisit the ER folding machinery. Here, we describe a reconstituted in-vitro system of purified human proteins that enabled us to delineate the function of TAPBPR during the UGGT1-catalyzed quality control and reglucosylation of MHC I. By combining glycoengineering with liquid chromatography-mass spectrometry, we show that TAPBPR promotes reglucosylation of peptide-free MHC I by UGGT1. Thus, UGGT1 cooperates with TAPBPR in fulfilling a crucial function in the quality control mechanisms of antigen processing and presentation.

## Editor's evaluation

This valuable study reports a complete in-vitro system where different steps and direct interactions between different components of MHC I maturation can be monitored, hence leading to a better mechanistic understanding of MHC I maturation. The evidence supporting the findings is solid and the methods, data and analyses broadly support the claims with only minor weaknesses. This work will be of interest to immunologists and biochemists.

*For correspondence: tampe@em.uni-frankfurt.de

Competing interest: The authors declare that no competing interests exist.

## Introduction

The adaptive immune system is able to detect diseased cells by scanning cell-surface-presented major histocompatibility complex class I (MHC I) molecules loaded with peptides (*Blum et al., 2013*; *Pishesha et al., 2022*; *Rock et al., 2016*). The peptidic ligands of these peptide-MHC I (pMHC I) complexes are primarily derived from proteasomal degradation products of cytosolic proteins and are transported into the endoplasmic reticulum (ER) by the ATP-binding cassette (ABC) transporter associated with antigen processing (TAP1/2) (*Thomas and Tampé, 2020*; *Thomas and Tampé, 2021*). Inside the ER, peptides can be further trimmed by aminopeptidases, in particular ERAP1/2, generating ligands of optimal length for fitting in the MHC I binding groove (*Hammer et al., 2007*; *Hattori and Tsujimoto, 2013*). The antigen transporter TAP1/2 is the central component of a dynamic supramolecular machinery called the peptide-loading complex (PLC) coordinating peptide transfer across the ER membrane with peptide loading onto MHC I inside the ER lumen (*Blees et al., 2017*; *Trowitzsch and Tampé, 2020*). Peptide loading and optimization is facilitated by the MHC I-specific chaperones tapasin (Tsn) and TAPBPR (TAP binding protein related; *Margulies et al., 2022*; *Thomas and Tampé,*

*2021*). Both Tsn and TAPBPR stabilize intrinsically unstable peptide-deficient MHC I and act as peptide editors or proofreaders by accelerating peptide exchange and selecting high-affinity peptide ligands (*Boyle et al., 2013*; *Chen and Bouvier, 2007*; *Fleischmann et al., 2015*; *Hermann et al., 2013*; *Hermann et al., 2015*; *Morozov et al., 2016*; *Sagert et al., 2020*; *Tan et al., 2002*; *Wearsch and Cresswell, 2007*). The affinity optimization of MHC I-associated peptides is crucial for immunosurveillance by T lymphocytes, priming of naive T cells, and T cell differentiation. The catalytic principles of peptide optimization have been revealed by crystal structures of the TAPBPR-MHC I complex (*Jiang et al., 2017*; *Thomas and Tampé, 2017*). Insights into the interaction characteristics between Tsn and peptide-receptive MHC I have been gained by structures of Tsn-ERp57-H2-D$^b$ (*Müller et al., 2022*) and Tsn-HLA-B*44:05 (*Jiang et al., 2022*), respectively. While Tsn is a key constituent of the PLC, TAPBPR functions outside the PLC and downstream of Tsn in the secretory pathway, even beyond the ER in the *Golgi* compartment (*Boyle et al., 2013*).

As glycoproteins consisting of a highly polymorphic, N-glycosylated heavy chain and the invariant light chain β2-microglobulin (β2m), MHC I complexes are also subject to quality control by the calnexin/calreticulin cycle in which the major glycoprotein folding sensor UDP-glucose:glycoprotein glucosyltransferase 1 (UGGT1) fulfills a crucial role (*Wearsch et al., 2011*; *Zhang et al., 2011*): The two outermost glucose residues of the core N-glycan $Glc_3Man_9GlcNAc_2$, which is co-translationally transferred to a conserved asparagine of the nascent MHC I heavy chain, are sequentially cleaved off by α-glucosidase I and II (GluI/II), and the resulting mono-glucosylated dodecasaccharide ($Glc_1Man_9GlcNAc_2$) is specifically recognized by the scaffolding lectin-like ER chaperones calnexin and calreticulin. The latter escorts MHC I to the PLC for peptide loading. Once pMHC I complexes are no longer protected by calreticulin, GluII can cleave off the terminal glucose (*Domnick et al., 2022*). Those pMHC I complexes that have acquired optimal peptides via the PLC or in a PLC-independent fashion and have adopted their native fold when their last glucose residue is removed, can exit the ER and travel via the *Golgi* apparatus to the cell surface where they are scanned by T cell receptors (TCRs) on cytotoxic T cells and natural killer (NK) cell receptors. However, the $Man_9GlcNAc_2$ glycan of MHC I complexes that are suboptimally loaded or empty and partially misfolded is recognized and reglucosylated by UGGT1, allowing the newly formed mono-glucosylated MHC I species to be re-engaged by calnexin or calreticulin and given another chance to acquire a suitable peptide, for example, by revisiting the PLC (*Wearsch et al., 2011*; *Zhang et al., 2011*).

Most interestingly, the UGGT1-mediated quality control of MHC I has recently been shown to be directly promoted by the peptide editor TAPBPR, based on a combination of immunoprecipitation experiments from cell lysates and mass spectrometry (*Neerincx et al., 2017*). Here, we aimed to characterize the cooperation and allosteric crosstalk between UGGT1 and TAPBPR in the quality control of human MHC I, using purified components under defined reaction conditions. We describe an in-vitro reglucosylation assay that is based on a combination of protein glycoengineering and top-down liquid chromatography-mass spectrometry (LC-MS) that enabled us to show that TAPBPR is both necessary and sufficient to mediate reglucosylation of the human MHC I allomorph HLA-A*68:02 by UGGT1. We demonstrate a direct interaction between TAPBPR-MHC I and UGGT1 using purified human proteins. Our findings corroborate the notion that TAPBPR unmediatedly aids UGGT1 in reglucosylating peptide-deficient MHC I, thereby affecting the maturation and quality control of MHC I in antigen processing and presentation beyond its peptide selector function.

## Results
### Design of an in-vitro system of UGGT1-catalyzed quality control

To investigate the role of TAPBPR in the UGGT1-catalyzed quality control of MHC I, we developed an in-vitro reglucosylation system consisting of human UGGT1 and peptide-receptive MHC I-TAPBPR complexes containing the ER-lumenal domains of TAPBPR and the MHC I allomorph HLA-A*68:02. HLA-A*68:02 is a favored client of TAPBPR (*Boyle et al., 2013*). The conditions of this experimental system mimicked the peptide-depleted physiological environments TAPBPR is operating in. Constructs for co-expression and secretion of TAPBPR and HLA-A*68:02 from HEK293-F cells were designed as previously described (*O'Rourke et al., 2019*). In short, HLA-A*68:02 was expressed as a fusion between β2m and heavy chain, with a cleavable C-terminal Fos leucine zipper domain (*Figure 1A*), whereas the TAPBPR construct contained a cleavable C-terminal Jun leucine zipper domain and a

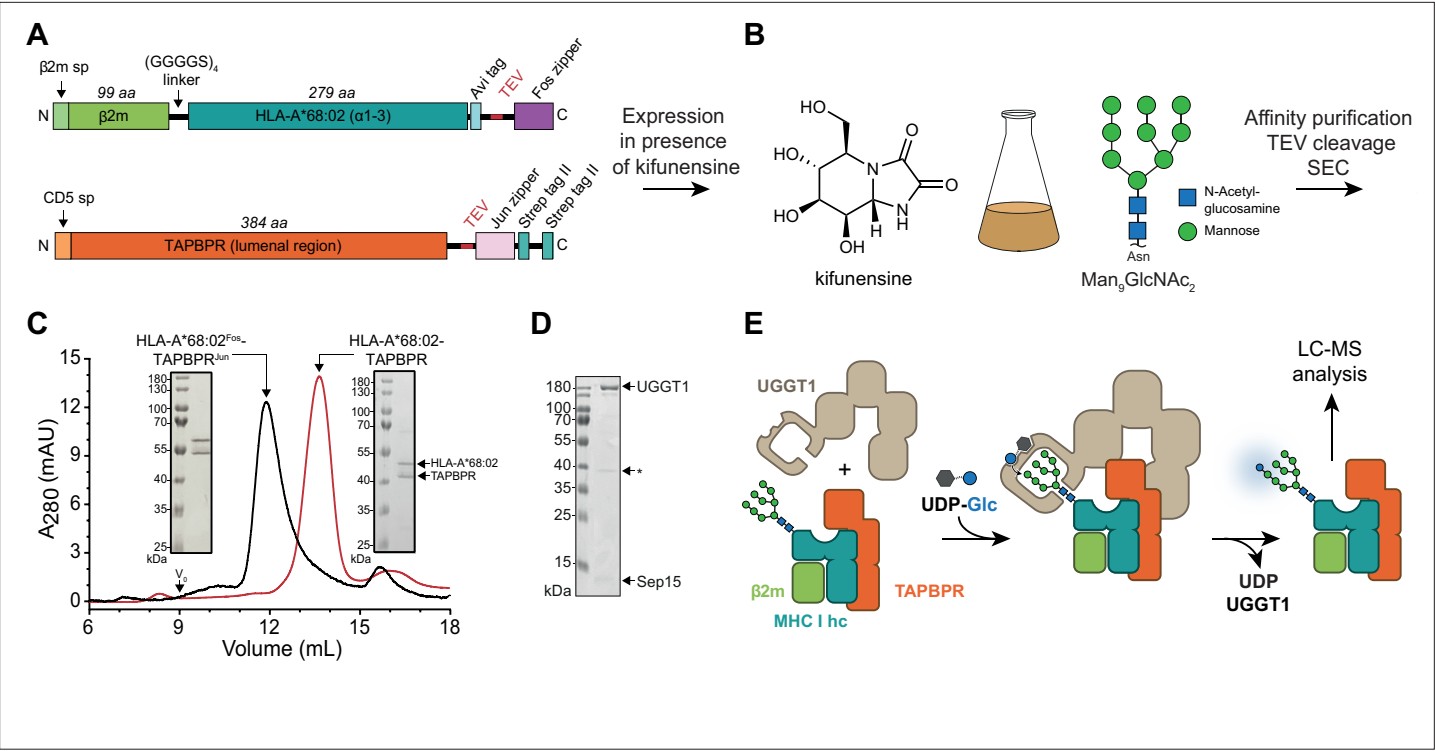

**Figure 1.** In-vitro system of UGGT1-catalyzed quality control and reglucosylation of MHC I. (**A**) Construct design to obtain peptide-receptive HLA-A*68:02-TAPBPR complexes by transient co-transfection of HEK293-F cells. (**B**) Co-expression of HLA-A*68:02 and TAPBPR was performed in the presence of the α-mannosidase I inhibitor kifunensine to generate a defined $Man_9GlcNAc_2$ glycan tree on the MHC I that can be recognized by UGGT1. (**C**) The secreted leucine zippered complex was isolated from the cell culture supernatant by Strep-Tactin affinity purification. Leucine zippers and the Twin-Strep-tag of TAPBPR were removed by protease treatment. The MHC I-chaperone complexes were further analyzed and purified by size exclusion chromatography (SEC) on a Superdex 200 (Increase 10/300) column. (**D**) SDS-PAGE analysis of immobilized metal-affinity chromatography (IMAC)-purified human wildtype (wt) UGGT1 co-expressed with Sep15 and secreted from insect cells. The asterisk (*) indicates a degradation product of UGGT1. (**E**) The purified UGGT1-Sep15 complex was employed in the reglucosylation assay with the peptide-receptive HLA-A*68:02-TAPBPR complex of (**C**) harboring the high-mannose glycan on MHC I. Abbreviations: $A_{280}$: absorption at 280 nm; aa: amino acids; kDa: kilodalton; mAU: milli-absorption units; MHC I hc: MHC I heavy chain; sp: signal peptide; UDP-Glc: UDP-glucose; $V_0$: void volume.

The online version of this article includes the following source data and figure supplement(s) for figure 1:

**Source data 1.** Original SDS-PAGE gel of HLA-A*68:02^Fos^-TAPBPR^Jun^, **Figure 1C**.

**Source data 2.** Original SDS-PAGE gel of HLA-A*68:02-TAPBPR, **Figure 1C**.

**Source data 3.** Original SDS-PAGE gel of UGGT1-Sep15, **Figure 1D**.

**Figure supplement 1.** Influence of kifunensine concentration on the proportion of $Man_9GlcNAc_2$ among the $Man_{7-9}GlcNAc_2$ MHC I glycan species when co-expressed with TAPBPR in HEK293-F cells.

**Figure supplement 2.** Analysis of purified human UGGT1.

**Figure supplement 2—source data 1.** Original anti-UGGT1 immunoblot, **Figure 1—figure supplement 2A**.

**Figure supplement 2—source data 2.** Original SDS-PAGE gel of UGGT1^D1316N^, **Figure 1—figure supplement 2C**.

Twin-Strep-tag (**Figure 1A**). Co-expression of MHC I with TAPBPR was carried out in the presence of the alkaloid kifunensine to obtain the HLA-A*68:02 species carrying at Asn86 the $Man_9GlcNAc_2$ glycan that is recognized by UGGT1 (**Figure 1B**). Kifunensine is an α-mannosidase I inhibitor that arrests glycan processing primarily at the $Man_9GlcNAc_2$ stage (**Chang et al., 2007**; **Elbein, 1991**). We examined different kifunensine concentrations to optimize the proportion of $Man_9GlcNAc_2$ among the $Man_{7-9}GlcNAc_2$ MHC I glycan species, as determined by LC-MS. The highest yield of $Man_9GlcNAc_2$ was achieved with 10 µM kifunensine, while increasing the concentration to 20 µM did not improve the yield (**Figure 1—figure supplement 1**). After Strep-Tactin affinity purification and *Tobacco etch* virus (TEV) protease cleavage, the peptide-receptive HLA-A*68:02-TAPBPR complex eluted as monodisperse sample at the expected size of 90 kDa during size-exclusion chromatography (SEC) (**Figure 1C**).

His-tagged human UGGT1 was co-expressed and secreted from insect cells with its binding partner Sep15 (15 kDa selenoprotein), which has been shown to enhance UGGT1 activity (*Takeda et al., 2014*). The UGGT1-Sep15 complex (hereafter referred to simply as UGGT1) was purified from the cell culture supernatant by immobilized-metal affinity chromatography (IMAC) via the C-terminal histidine tag of the UGGT1 construct (*Figure 1D* and *Figure 1—figure supplement 2*). In addition to full-length UGGT1, we detected small amounts of a degradation product at ~37 kDa by SDS-PAGE and immunoblotting, representing the C-terminal catalytic glucosyltransferase domain, in line with previous observations (*Guerin and Parodi, 2003*; *Figure 1D* and *Figure 1—figure supplement 2A*). To analyze UGGT1-mediated quality control of MHC I in our in-vitro reglucosylation assay, we started the reaction by adding purified UGGT1 together with UDP-glucose in $Ca^{2+}$-containing buffer to the purified peptide-receptive HLA-A*68:02-TAPBPR complex (*Figure 1E*). LC-MS analysis allowed us to monitor the progress of the reglucosylation reaction on the TAPBPR-chaperoned HLA-A*68:02.

## UGGT1 catalyzes the reglucosylation of TAPBPR-bound HLA-A*68:02

Our reglucosylation assay allowed us to investigate the activity of UGGT1 towards TAPBPR-bound HLA-A*68:02 harboring the $Man_9GlcNAc_2$ glycan. UGGT1 exhibited clear UDP-glucose-dependent glucosyltransferase activity towards $Man_9GlcNAc_2$-HLA-A*68:02-TAPBPR, as judged by the appearance of the peak corresponding to the HLA-A*68:02-linked mono-glucosylated $Glc_1Man_9GlcNAc_2$ glycan species in the mass spectra (*Figure 2A–C*). The D1316N mutant of UGGT1, in which the first aspartate of the $Ca^{2+}$-coordinating DxD motif in the catalytic site (*Roversi et al., 2017*) is mutated, was inactive (*Figure 2D*). Activity of mutant UGGT1^D1316N could not be restored by adding 5 µM purified Sep15 (*Figure 2D*). Interestingly, UGGT1-catalyzed reglucosylation reactions in general reached saturation at ~80% product formation (*Figure 2E*). This saturation could be overcome neither by raising the temperature (*Figure 2F*) nor by increasing the enzyme concentration (*Figure 2—figure supplement 1*).

## TAPBPR is indispensable for UGGT1-catalyzed reglucosylation of HLA-A*68:02

After having verified that our UGGT1 preparation is active towards the TAPBPR-chaperoned HLA-A*68:02 in its peptide-receptive state, we wondered whether the MHC I must be chaperoned by TAPBPR in order to be recognized as a substrate by UGGT1. Moreover, we wanted to scrutinize the importance of the peptide-loading status of the MHC I for the UGGT1-catalyzed reglucosylation reaction. To this end, we loaded HLA-A*68:02 with a photo-cleavable peptide by incubating the purified HLA-A*68:02-TAPBPR complex overnight with a 250-fold molar excess of the peptide (*Figure 3A*). Binding of high-affinity peptide leads to TAPBPR dissociation, allowing the peptide-loaded $Man_9GlcNAc_2$-HLA-A*68:02 to be isolated by size-exclusion chromatography (SEC; *Figure 3A and B*). The peptide-deficient state of the MHC I was generated by photo-cleavage of the bound peptide and dissociation of the resulting low-affinity peptide fragments. The peptide-loading status of pMHC I and empty MHC I was confirmed by LC-MS (*Figure 3C* and *Figure 3—figure supplement 1*). Additionally, the generated peptide-free $Man_9GlcNAc_2$-HLA-A*68:02 was shown to bind the conformation-specific antibody W6/32, demonstrating that the peptide-receptive MHC I remain stable throughout the experiment (*Figure 3—figure supplement 2*). Most strikingly, neither the peptide-loaded nor the peptide-deficient HLA-A*68:02 were reglucosylated by UGGT1 in the absence of TAPBPR (*Figure 3D–F*), but we were able to restore UGGT1 activity towards peptide-deficient HLA-A*68:02 by adding TAPBPR (*Figure 3F*). Moreover, loading of a low-affinity (suboptimal) peptide onto the peptide-receptive MHC I could not restore the susceptibility of UGGT1 (*Figure 3—figure supplement 3*), further emphasizing the critical role of TAPBPR. These results demonstrate that the MHC I-specific chaperone TAPBPR is an essential mediator in the UGGT1-catalyzed reglucosylation of the human MHC I allomorph HLA-A*68:02.

## Cys97 of TAPBPR is not required for interaction with UGGT1

The ER-lumenal part of the peptide proofreader TAPBPR consists of an N-terminal composite domain composed of coalesced seven-stranded β barrel and immunoglobulin (Ig)-like (V type) folds, and a C-terminal IgC1 domain (*Jiang et al., 2017*; *Thomas and Tampé, 2017*; *Figure 4A*). TAPBPR contacts the peptide-receptive MHC I client through a large concave surface of its N-terminal composite

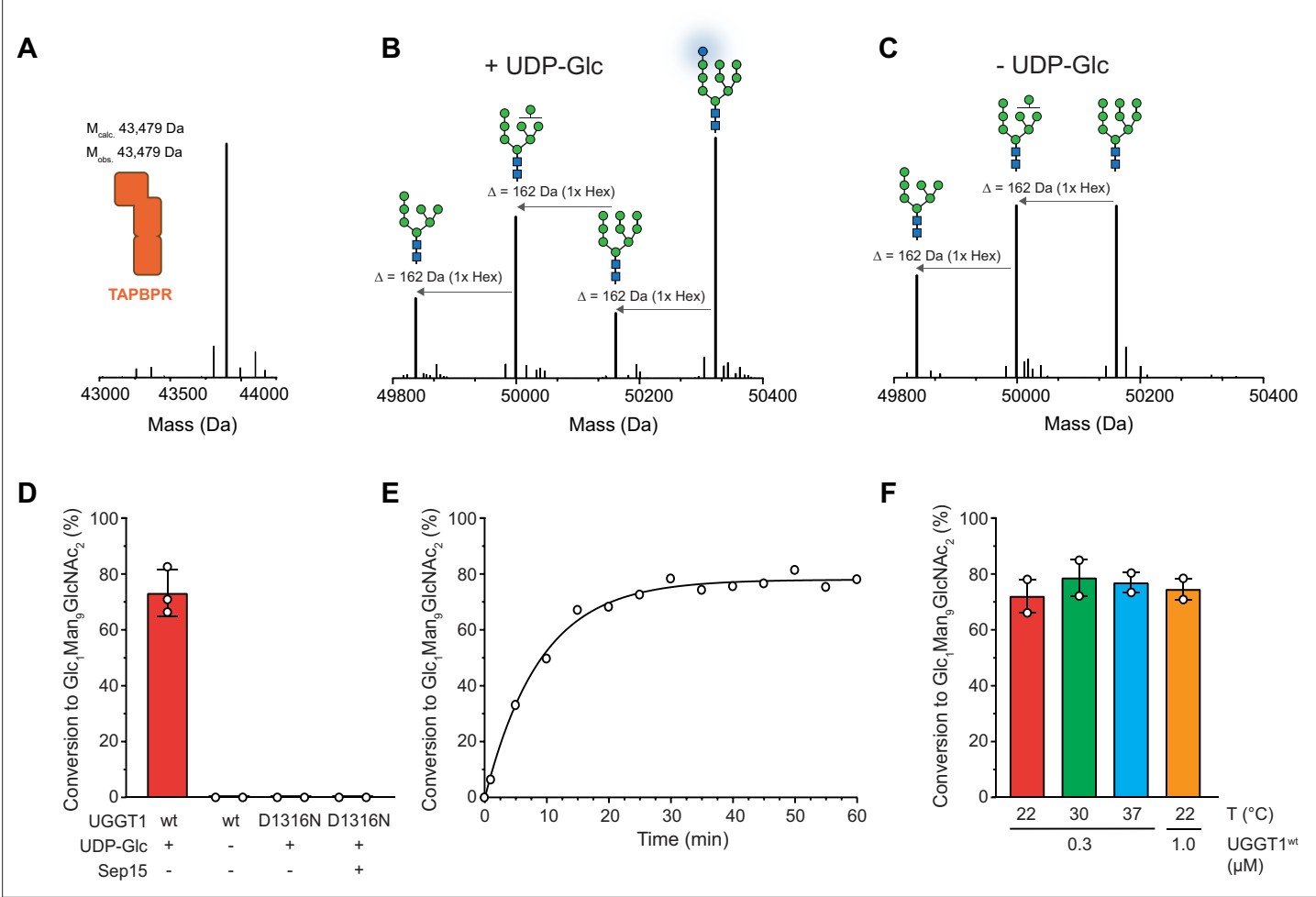

**Figure 2.** UGGT1 catalyzes reglucosylation of TAPBPR-bound HLA-A*68:02. Protein identities in the reconstituted glucosylation system and the glycan status of HLA-A*68:02 ($Man_{7.9}GlcNAc_2$ and $Glc_1Man_9GlcNAc_2$) during the UGGT1-catalyzed reglucosylation reaction were determined by liquid chromatography-mass spectrometry (LC-MS). (**A**) Deconvoluted mass spectrum of TAPBPR. (**B**) HLA-A*68:02-TAPBPR (3 µM) complex was incubated with UGGT1 (1 µM) in the absence (-) and (**C**) presence (+) of UDP-glucose (UDP-Glc) for 1 h at room temperature. In the presence of UDP-Glc, UGGT1 catalyzed the glucose transfer, generating the mono-glucosylated glycan ($Glc_1Man_9GlcNAc_2$), whereas the native glycan pattern remained unchanged in the absence of UDP-Glc. Exemplary deconvoluted mass spectra of glycosylated HLA-A*68:02 representing LC-MS analyses summarized in (**D**) are shown. (**D**) Comparison of glucosyltransferase activity, as measured by the amount of produced $Glc_1Man_9GlcNAc_2$-HLA-A*68:02, upon incubating HLA-A*68:02-TAPBPR (3 µM) with UGGT1 proteins (1 µM) in the absence (-) and presence (+) of UDP-Glc and additional purified Sep15 (5 µM), respectively. Data represent mean ± SD (n=3) and (n=2), respectively. (**E**) Kinetics of reglucosylation of HLA-A*68:02-TAPBPR (3 µM) catalyzed by UGGT1 (1 µM). (**F**) Temperature dependence of UGGT1-catalyzed reglucosylation of HLA-A*68:02-TAPBPR (3 µM). Reactions were stopped after 60 min. Data represent mean ± SEM (n=2). Abbreviations: Da: dalton; Glc: glucose; GlcNAc: N-acetylglucosamine; Hex: hexose; Man: mannose; $M_{calc}$: calculated mass; $M_{obs}$: observed mass.

The online version of this article includes the following figure supplement(s) for figure 2:

**Figure supplement 1.** UGGT1 concentration-dependent reglucoslyation kinetics of TAPBPR-bound HLA-A*68:02.

domain, which interacts mainly with the α2–1-helix region of the MHC I, and via interfaces of its C-terminal IgC1 domain with β2m and the α3 domain of the MHC I heavy chain (*Jiang et al., 2017*; *Thomas and Tampé, 2017*). Of the seven cysteines in the ER-lumenal portion of TAPBPR, four are involved in two disulfide bridges in the N-terminal domain, and two form a disulfide bond in the C-terminal domain (*Jiang et al., 2017*; *Thomas and Tampé, 2017*; *Figure 4A*). The seventh cysteine residue, C97, is surface-exposed and located opposite to the concave surface in the N-terminal domain (*Thomas and Tampé, 2017*; *Figure 4A*). Mutating C97 to alanine was shown to have no effect on the catalytic activity of TAPBPR (*Neerincx et al., 2017*). With respect to UGGT1 binding, C97 is not engaged in an intermolecular disulfide bond with UGGT1, but seems necessary for the interaction with UGGT1 in

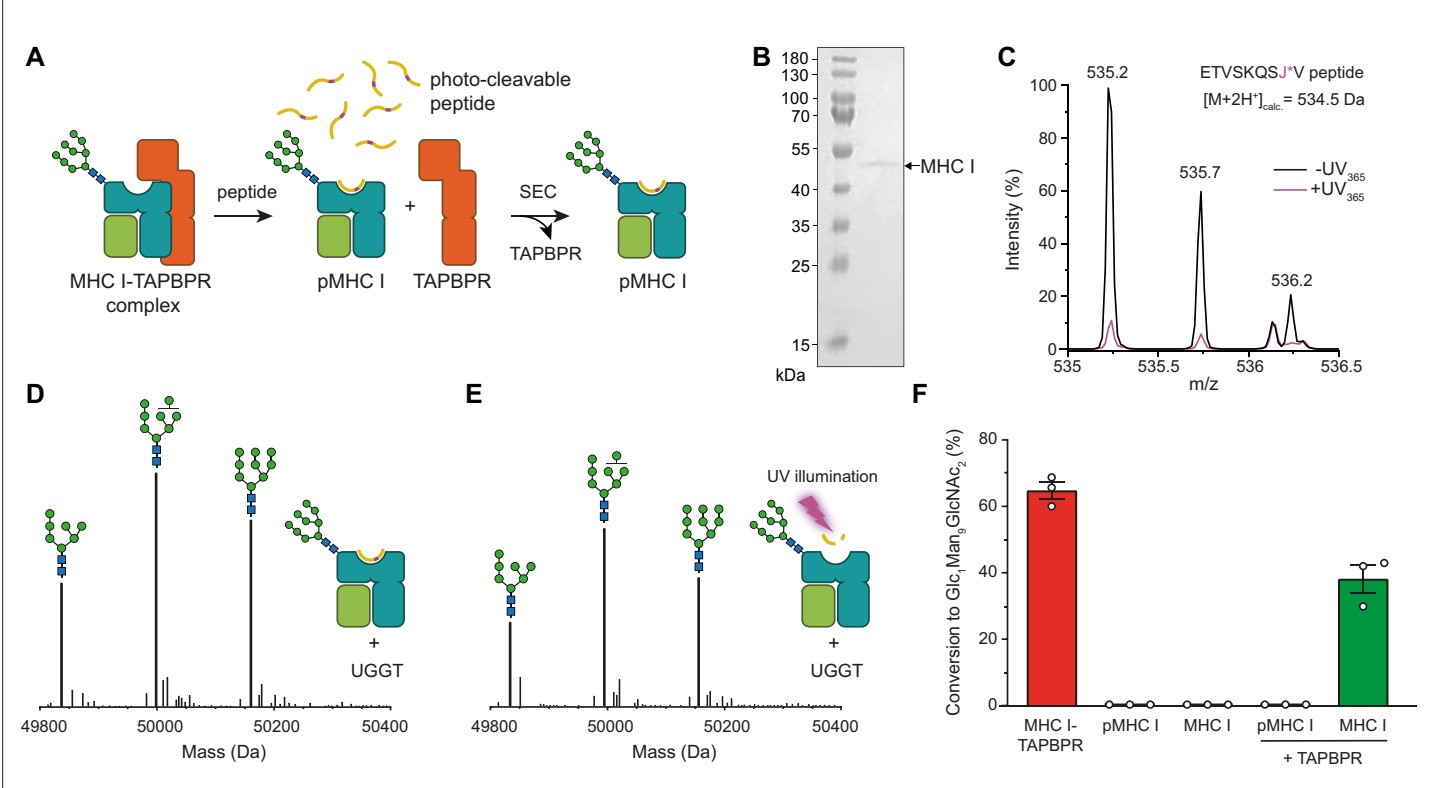

**Figure 3.** TAPBPR is indispensable for UGGT1-catalyzed reglucosylation of HLA-A*68:02. (**A**) Schematic representation of loading a photo-cleavable peptide onto $Man_9GlcNAc_2$-HLA-A*68:02 via pre-assembled MHC I-TAPBPR complexes. Peptide binding results in TAPBPR dissociation, permitting isolation of photo-conditional pMHC I complexes. (**B**) Non-reducing SDS-PAGE analysis of SEC-isolated pMHC I. (**C**) Isotope-resolved raw mass spectrum of the double-charged photo-cleavable peptide ETVSKQSJ*V (J*, 3-amino-3-(2-nitrophenyl)-propanoic acid) bound to HLA-A*68:02 before (black) and after (violet) UV illumination at 365 nm. (**D**) Reglucosylation experiment of peptide-loaded and (**E**) peptide-free HLA-A*68:02 (1 µM) in the absence of TAPBPR. $Man_9GlcNAc_2$-HLA-A*68:02 devoid of peptide was produced by photo-cleavage and release of peptide fragments prior to incubation with UGGT1 (600 nM). Shown are representative LC-MS analyses of data depicted in F. (**F**) Reglucosylation activity of UGGT1 (600 nM) towards TAPBPR-bound HLA-A*68:02 (1 µM), peptide-bound HLA-A*68:02 (pMHC I, 1 µM), and UV-illuminated HLA-A*68:02 lacking peptide (1 µM). If TAPBPR (6 µM) is added back to peptide-deficient MHC I (1 µM), recognition of MHC I as a substrate by UGGT1 is restored. The amount of produced $Glc_1Man_9GlcNAc_2$ glycan was determined after 60 min for all reactions. Data represent mean ± SEM (n=3). Abbreviations: calc.: calculated; Da: dalton; kDa: kilodalton; SEC: size-exclusion chromatography.

The online version of this article includes the following source data and figure supplement(s) for figure 3:

**Source data 1.** Original SDS-PAGE gel of SEC isolated HLA-A*68:02, *Figure 3B*.

**Figure supplement 1.** SEC-MS analysis of photo-cleavable peptide bound to $Man_9GlcNAc_2$-HLA-A*68:02 before and after UV illumination.

**Figure supplement 2.** Conformation-specific antibody W6/32 binds to peptide-loaded and peptide-receptive β2m-HLA-A*68:02.

**Figure supplement 3.** UGGT1-mediated reglucosylation is independent of the MHC I peptide-loading status.

immunoprecipitates (*Neerincx et al., 2017*). In addition, HeLaM cells expressing TAPBPR[C97A] instead of TAPBPR[wt] displayed an altered peptide repertoire with a decrease in canonical anchor residues for HLA-A*68:02 and an increasing amount of peptide-deficient HLA-A*68:02 (*Neerincx et al., 2017*).

To explore the role of C97 in our in-vitro system of TAPBPR-mediated quality control of MHC I by UGGT1, we tested $Man_9GlcNAc_2$-HLA-A*68:02 bound to the TAPBPR[C97A] mutant for reglucosylation. To our surprise, UGGT1 exhibited consistently and significantly higher reglucosylation activity towards the mutant MHC I-TAPBPR[C97A] complex than towards the wildtype complex (*Figure 4B*). To exclude that the effect is caused by potentially pre-bound peptides of the purified MHC I-TAPBPR complexes, we analyzed the peptide-binding competence of the tethered complexes. Unless the Jun-Fos zippers have been removed, neither low- nor high-affinity peptides could bind to the complexes (*Figure 4—figure supplement 1*), verifying that the observed differences in reglucosylation are not caused by different peptides. Furthermore, in a pull-down experiment with UGGT1 (3 µM) using

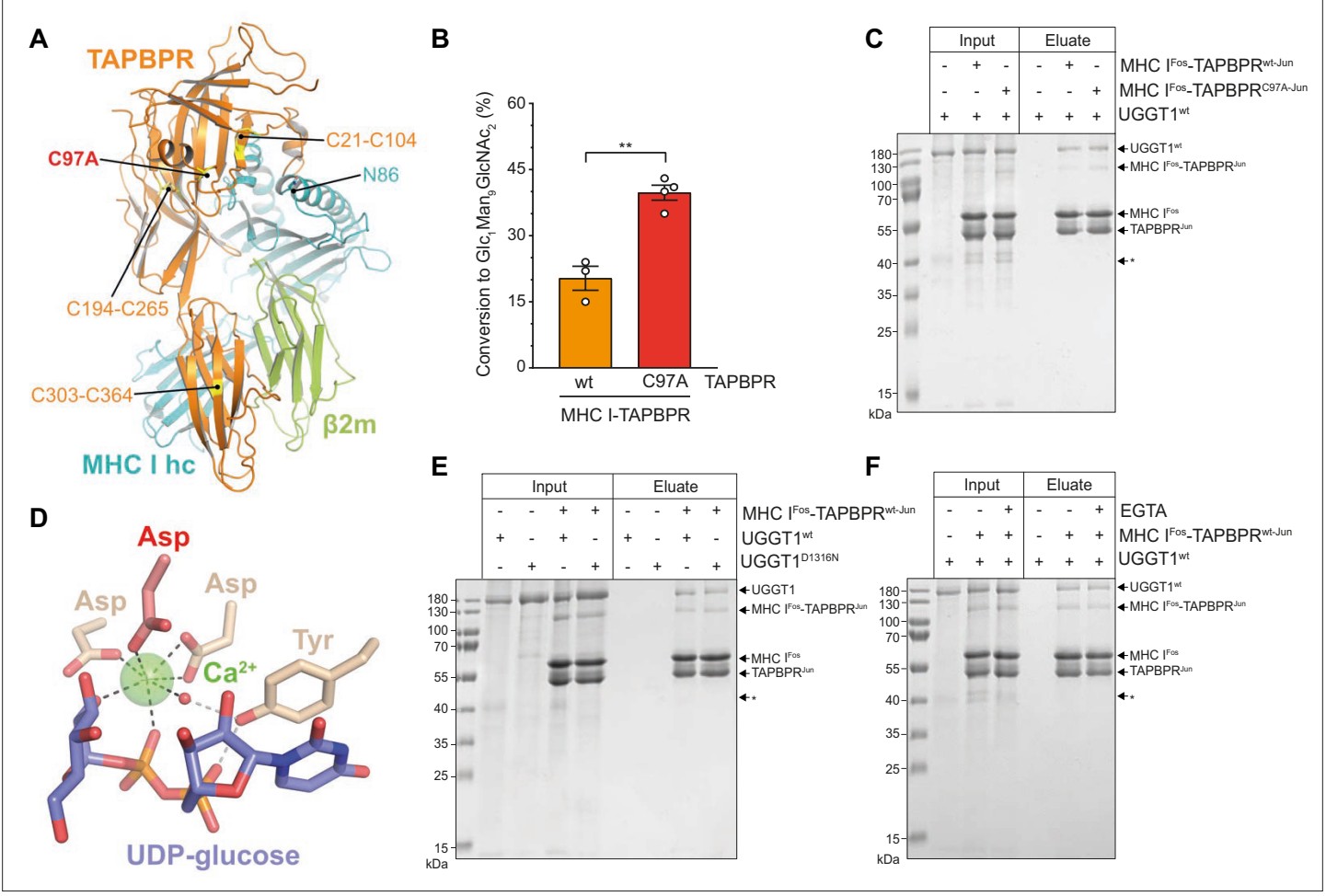

**Figure 4.** The interaction between UGGT1 and TAPBPR is independent of Cys97 and $Ca^{2+}$. (**A**) Cartoon representation of the MHC I-TAPBPR complex (PDB ID: 5opi). Disulfide bridges in TAPBPR and C97, which was mutated to alanine, are shown as sticks and highlighted in yellow. The glycosylated N86 (depicted as sticks) of the MHC I heavy chain (hc) lies in the vicinity of C97. (**B**) UGGT1wt-catalyzed (1 µM) reglucosylation of $Man_9GlcNAc_2$-HLA-A*68:02 bound to TAPBPRwt (3 µM) or TAPBPRC97A (3 µM). Data represent mean ± SD (n=3 and 4). Statistical analysis was performed using unpaired t-test. Asterisks indicate the level of significance (p-values): **p≤0.01. (**C**) Pull-down experiment with UGGT1wt (3 µM, present in each lane) using Twin-Strep-tagged $Man_9GlcNAc_2$-HLA-A*68:02-TAPBPR complexes (5 µM) captured on Strep-Tactin Sepharose. (**D**) $Ca^{2+}$ (green sphere) coordination in the active site of the *Thermomyces dupontii* UGGT glucosyltransferase domain (PDB ID: 5h18). The side chain of the corresponding aspartate residue that has been mutated to asparagine in human UGGT1D1316N (see panel (**F**)) is shown in red. The red sphere represents a water molecule that occupies one of the coordination sites. (**E**) Pull-down experiment with UGGT1wt (5 µM) and the catalytically inactive mutant UGGT1D1316N (5 µM) using Twin-Strep-tagged $Man_9GlcNAc_2$-HLA-A*68:02-TAPBPRwt complex (3 µM) in the presence of 5 mM $CaCl_2$. Eluates of the pull-down experiments were analyzed by SDS-PAGE (12.5%, non-reducing, Coomassie-stained). (**F**) $Ca^{2+}$ dependence of UGGT1wt (3 µM) binding to $Man_9GlcNAc_2$-HLA-A*68:02-TAPBPRwt (5 µM) was analyzed in presence (+) and absence (-) of EGTA. The asterisk (*) indicates a degradation product of UGGT1, representing the glycosyltransferase domain. The data are representative of two independent experiments. Abbreviation: kDa: kilodalton. The numbering of TAPBPR refers to the mature protein as defined by N-terminal sequencing (*Zhang and Henzel, 2004*).

The online version of this article includes the following source data and figure supplement(s) for figure 4:

**Source data 1.** Original SDS-PAGE gel of pull-down experiment with UGGT1wt, *Figure 4C*.

**Source data 2.** Original SDS-PAGE gel of pull-down experiment with UGGT1wt and UGGT1D1316N, *Figure 4E*.

**Source data 3.** Original SDS-PAGE gel of pull-down experiment to test $Ca^{2+}$ dependence of UGGT1wt, *Figure 4F*.

**Figure supplement 1.** Tethered MHC IFos-TAPBPRJun complexes are not competent in peptide-binding.

$Man_9GlcNAc_2$-HLA-A*68:02 bound to Twin-Strep-tagged TAPBPR (5 µM) and captured on Strep-Tactin Sepharose, we observed that the interaction between UGGT1 and HLA-A*68:02-TAPBPR was not affected by the C97A mutation (*Figure 4C*). We also noticed that in the eluate of this pull-down experiment, the degradation product at ~37 kDa representing the C-terminal catalytic glucosyltransferase

domain of UGGT1 was not present, indicating that the interface of UGGT1 with HLA-A*68:02-TAPBPR extends beyond the glucosyltransferase domain (*Figure 4C*, asterisk).

UGGT1 requires calcium ions for its glucosyltransferase activity (*Trombetta and Parodi, 1992*). $Ca^{2+}$ is coordinated by the DxD motif in the catalytic site that is typical of most A-type glycosyltransferases (*Lairson et al., 2008*; *Roversi et al., 2017*; *Satoh et al., 2017*; *Figure 4D*). The interaction with the $Man_9GlcNAc_2$-MHC I-TAPBPR complex was not affected by the active-site mutation in $UGGT1^{D1316N}$ which disrupts the $Ca^{2+}$-coordinating DxD motif (*Figure 4E*). To test whether $Ca^{2+}$ influences the interaction between UGGT1 and $Man_9GlcNAc_2$-MHC I-TAPBPR in the absence of the donor substrate UDP-glucose, we performed the Strep-Tactin-based pull-down experiment in the absence and presence of the $Ca^{2+}$ chelator EGTA. We found no difference in the amount of UGGT1 that was pulled down by HLA-A*68:02-TAPBPR between the two different conditions (*Figure 4F*), suggesting that the binding of UGGT1 to TAPBPR-associated $Man_9GlcNAc_2$-HLA-A*68:02 is $Ca^{2+}$-independent.

## Discussion

UGGT1 is a pivotal component of the multi-chaperone machinery constituting the ER quality-control network in eukaryotes. The physiological significance of this enzyme is illustrated by the fact that deletion of its gene in mice is embryonically lethal (*Molinari et al., 2005*). UGGT1 functions as the major eukaryotic glycoprotein folding sensor and gatekeeper in the calnexin/calreticulin cycle, preventing deglucosylated, non-natively folded N-glycosylated proteins from leaving the ER (*Adams et al., 2020*; *Caramelo and Parodi, 2015*). Gaining insights into how UGGT1 recognizes its client proteins and feeds them back into the cycle of the lectin-like scaffolding chaperones calnexin and calreticulin is therefore of fundamental importance for a deeper understanding of cellular proteostasis. The catalytic activity of UGGT1 has so far been studied by employing a small set of specially prepared model substrates (*Ritter and Helenius, 2000*; *Ritter et al., 2005*), glycopeptides (*Taylor et al., 2003*), or synthetic substrates (*Kiuchi et al., 2018*; *Totani et al., 2005*; *Totani et al., 2009*).

Here, we designed an experimental in-vitro approach to investigate UGGT1-catalyzed quality control and reglucosylation of MHC I, a physiological client of outstanding importance in human biology. Our strategy utilizes protein glycoengineering in human cells, taking advantage of the α-mannosidase I inhibitor kifunensine to generate $Man_9GlcNAc_2$-MHC I molecules. This approach is combined with LC-MS, which allows the UGGT1-catalyzed glucosylation reaction to be monitored (*Figure 1*). Using our reglucosylation system, we were able to successfully reconstruct the cellular process of UGGT1-mediated quality control of MHC I in-vitro with purified protein components (*Figure 2*). Under the chosen conditions, the UGGT1-catalyzed conversion of $Man_9GlcNAc_2$-MHC I to $Glc_1Man_9GlcNAc_2$-MHC I reached saturation at about 80%, similar to previous observations with rat UGGT (*Totani et al., 2005*), and could not be increased by higher temperatures or higher enzyme concentrations (*Figure 2D–F*). Importantly, when analyzing the glucosylation reaction in the absence of TAPBPR on the isolated pMHC I charged with a high-affinity, photo-cleavable peptide, we observed no activity (*Figure 3D*). This is in agreement with the expectation that the quality-control sensor UGGT1 does not act on MHC I clients that are already loaded with optimal peptide cargo. This result highlights the ability of our reglucosylation assay to faithfully reconstruct in-vivo events of ER quality control. Most interestingly, when we converted the high-affinity peptide into dissociating low-affinity fragments by UV illumination, essentially generating a peptide-deficient, 'frustrated' MHC I which should meet the criteria of a *bona-fide* UGGT1 substrate, there was still no reglucosylation by UGGT1 detectable (*Figure 3E*). Only the addition of TAPBPR could restore UGGT1-catalyzed reglucosylation (*Figure 3F*). These findings demonstrate that TAPBPR is an essential mediator in the reglucosylation of HLA-A*68:02 by UGGT1. The strict dependence of UGGT1 on TAPBPR in acting towards HLA-A*68:02 is most likely not valid for all MHC I allomorphs, since the various allomorphs differ in their reliance on TAPBPR as a peptide editor. Indeed, experimental evidence suggests that UGGT1 can operate on MHC I in a TAPBPR-independent manner (*Neerincx et al., 2017*), and it appears that UGGT1 interacts with most HC10-reactive MHC I allomorphs independent of TAPBPR, whereas the binding of W6/32-reactive MHC I allomorphs is TAPBPR-dependent (*Neerincx et al., 2017*). Notably, previous studies reported a preferred glucosylation by *Drosophila* UGGT1 towards the human allomorph HLA-B*08:01 when loaded with suboptimal peptides (*Wearsch et al., 2011*; *Zhang et al., 2011*). However, both studies applied MHC I complexes expressed in insect cells but glycan modification and a potential role of TAPBPR were not further addressed. In addition, it has been demonstrated that HLA-B*08:01

displays no detectable interaction with human TAPBPR and thus, it seems likely that UGGT1 can act on some MHC I allomorphs in a TAPBPR-independent manner (*Ilca et al., 2019*; *Sun et al., 2023*). The strategic approach applied in this study combines protein glycoengineering in human cells with LC-MS analysis, providing the major advantage of a fully defined glycan pattern on HLA-A*68:02 which allowed us to monitor the UGGT1-catalyzed reglucosylation in detail.

The importance of C97, the sole free cysteine in TAPBPR, for the interaction with UGGT1 is called into question by our data. While a previous study concluded that C97 is essential for the association between TAPBPR and UGGT1 (*Neerincx et al., 2017*), we found that the mutation of C97 to alanine affected neither the catalyzed reglucosylation of MHC I-TAPBPR$^{C97A}$ nor the interaction with the glucosyltransferase (*Figure 4B and C*). According to one of the TAPBPR-MHC I crystal structures (*Thomas and Tampé, 2017*), C97 is an exposed surface residue positioned opposite the concave surface in the N-terminal domain that engulfs the α2–1 region of the MHC I, relatively close to the glycosylated N86 in the MHC I (Cα-Cα distance: 23.5 Å; *Figure 4A*). While we do not exclude that C97 is part of the interface with UGGT1, we have no experimental evidence that its mutation has any negative effects on complex formation.

Despite Ca$^{2+}$ being an essential cofactor for UGGT1 to function as a catalyst of the glucosyl transfer reaction, we reveal that the metal ion is not required for the recognition of Man$_9$GlcNAc$_2$-MHC I-TAPBPR as substrate. Neither the mutation of the Ca$^{2+}$-coordinating DxD sequence motif in UGGT1 nor the presence of the Ca$^{2+}$ chelator EGTA changed the binding of the glucosyltransferase to Man$_9$GlcNAc$_2$-MHC I-TAPBPR (*Figure 4E and F*).

Taken together, the data we present here add to our current picture of MHC I quality control inside the ER, where UGGT1 and TAPBPR emerge as team players, with TAPBPR tracking down and stabilizing empty or suboptimally loaded MHC I complexes. For those MHC I complexes that have been deglucosylated by GluII and cannot be loaded with optimal peptide due to scarcity of suitable cargo, TAPBPR acts as a mediator for UGGT1-catalyzed reglucosylation. This appears to apply to at least a large subset of HLA-A allomorphs, many of which obtain their peptide in a Tsn-independent manner (*Greenwood et al., 1994*). Thus, the TAPBPR-UGGT1 complex promotes recycling of immature MHC I to the Glc$_1$Man$_9$GlcNAc$_2$-specific calnexin/calreticulin cycle and/or to the second peptide editor tapasin inside the PLC via calreticulin (*Neerincx et al., 2017*; *Wearsch et al., 2011*). Exactly how the handover from one MHC I chaperone to the other via calreticulin is taking place is an important question and remains to be investigated.

Future studies of TAPBPR-UGGT1-mediated MHC I quality control will need to decipher the detailed molecular mechanisms of this fundamental process to understand how TAPBPR binds UGGT1 and how UGGT1 interacts with MHC I clients. Is C97 of TAPBPR involved in the interface with UGGT1, and how can the seemingly contradictory findings concerning its significance for the interaction be explained? Does UGGT1 recognize only the MHC I glycan or maybe also features of the peptide-receptive conformation adopted by the MHC when bound to TAPBPR? Answers to these questions would most likely be provided by a structure of the MHC I-TAPBPR-UGGT1 complex captured *in flagranti* before the reglucosylation reaction happens. In terms of functional studies, we are confident that the in-vitro system described here will help dissect the synergistic interplay between ER chaperones in shepherding and assisting immunity-related glycoproteins and the N-glycoproteome in general along their maturation pathways and in cellular proteostasis.

## Materials and methods

**Key resources table**

| Reagent type (species) or resource | Designation | Source or reference | Identifiers | Additional information |
|---|---|---|---|---|
| Gene (human) | TAPBPR$^{wt-Jun}$ | *O'Rourke et al., 2019* | | Lumenal region |
| Gene (human) | TAPBPR$^{C97A-Jun}$ | This study | | Lumenal region, TAPBPR version of TAPBPR$^{wt-Jun}$ (see material and methods, DNA constructs) |
| Gene (human) | β2m-HLA-A*68:02 | *O'Rourke et al., 2019* | | Lumenal region |

*Continued on next page*

*Continued*

| Reagent type (species) or resource | Designation | Source or reference | Identifiers | Additional information |
|---|---|---|---|---|
| Gene (human) | UGGT1$^{wt}$-Sep15 | This study | | See Materials and methods, DNA constructs |
| Gene (human) | UGGT1$^{D1316N}$ | This study | | UGGT version of UGGT1$^{wt}$ (Dr. Pietro Roversi, Zitzmann lab, University of Oxford) |
| Gene (human) | Sep15 | This study | | See Materials and methods, DNA constructs |
| Strain, strain background (*Escherichia coli*) | BL21 (DE3) | Sigma-Aldrich | CMC0014 | Chemically competent |
| Cell line (Spodoptera Frugiperda) | *Sf*21 | Thermo Fisher Scientific | 11496015 | |
| Cell line (Human embryonic kidney) | HEK293-F | Thermo Fisher Scientific | R79007 | Routinely tested for mycoplasma contamination |
| Antibody | anti-human HLA-A/B/C (W6/32) (Mouse monoclonal) | BioLegend | Cat# 311402 RRID:AB_314871 | WB (1:3000) |
| Antibody | anti-UGGT/UGT1 (Rabbit monoclonal) | Abcam | Cat# ab124879 RRID:AB_10971344 | WB (1:3000) |
| Recombinant DNA reagent | pFastBacI-gp67 | PMID:2905996 | | Transfer vector for Bac-to-Bac system |
| Recombinant DNA reagent | pcDNA3.1(+) | Thermo Fisher Scientific | V79020 | Mammalian expression vector |
| Recombinant DNA reagent | pOPINGTTGneo | Dr. Pietro Roversi, Zitzmann lab, University of Oxford | | Mammalian expression vector |
| Peptide, recombinant protein | ETVSKQSJ*V | Dr. Ines Katharina Müller, Tampé lab | | J* denotes photo-cleavable amino acid, see material and methods, peptide loading onto MHC I |
| Peptide, recombinant protein | ILKCLEEPSV | This study | | See Materials and methods, UGGT1-mediated reglucosylation |
| Chemical compound, drug | UDP-Glc | Sigma-Aldrich | U4625 | |
| Chemical compound, drug | Kifunensine | BioMol | 10009437 | |
| Chemical compound, drug | VPA | Merck Millipore | P4543 | |
| Chemical compound, drug | Polyethyleneimine, linear | Sigma-Aldrich | 765090 | |
| Software, algorithm | OriginPro 2022b | | | |
| Software, algorithm | Prism 10 | GraphPad Software | | |
| Software, algorithm | Unify 1.9.4.053 | Walters | | |
| Other | Superdex 200 Increase 10/300 | GE Healthcare | 28990944 | SEC column, see Materials and methods, Purification of MHC I$^{Fos}$-TAPBPR$^{Jun}$ complexes |
| Other | Superdex 200 Increase 3.2/300 | GE Healthcare | 28990946 | SEC column, see Materials and methods, peptide loading onto MHC I |
| Other | UPLC Peptide BEH C$_{18}$ Column | Walters | 186003555 | UPLC-column, see Materials and methods, LC-MS analysis |
| Other | UPLC Protein BEH C$_4$ Column | Walters | 186008471 | UPLC-column, see Materials and methods, LC-MS analysis |

*Continued on next page*

*Continued*

| Reagent type (species) or resource | Designation | Source or reference | Identifiers | Additional information |
|---|---|---|---|---|
| Other | Äkta Purifier | Cytiva | | Protein purification |
| Other | Äkta Ettan | GE Healthcare | | Protein purification |
| Other | BioAccord LC-MS system | Walters | | See Materials and methods, LC-MS analysis |
| Other | Tabacco Etch Virus | In house production | | TEV protease cleavage |

## DNA constructs

The previously described codon-optimized DNA constructs of single-chain (sc) β2m-HLA-A*68:02[Fos] and TAPBPR[Jun] (*O'Rourke et al., 2019*) were ligated into pcDNA3.1. The mutant TAPBPR[C97A] was generated by site-directed mutagenesis using the following primers:
5'-TTTGCTCCACGCAGATGCCTCAGGCAAGGA-3' (forward)
5'-TCCTTGCCTGAGGCATCTGCGTGGAGCAAA-3' (reverse)

The numbering of TAPBPR refers to the mature protein as defined by N-terminal sequencing (*Zhang and Henzel, 2004*). The construct of human UGGT1[wt] in the pOPINTTGneo vector containing a C-terminal His$_6$-tag was kindly provided by Dr. Pietro Roversi of the Zitzmann lab (University of Oxford). UGGT1[D1316N] was produced by site-direct mutagenesis using the following primers:
5'-TGACAAGTTCCTGTTTGTGAATGCTGATCAGATTGTACG-3' (forward)
5'-CGTACAATCTGATCAGCATTCACAAACAGGAACTTGTCA-3' (reverse)

A customized version of the pFastBac vector containing the gp67 signal peptide was used to express human Sep15 in insect cells. For co-expression of human UGGT1 and Sep15 in insect cells, a customized version of the pFastBac Dual vector was used which contained the gp67 and melittin signal peptides. All constructs contained N-terminal signal peptides for secretion, and the selenocysteine of Sep15 was mutated to cysteine.

## Cell culture and transfection of HEK293 cells

HEK293-F cells were grown in FreeStyle 293 Expression Medium (Gibco) at 37 °C, 5% CO$_2$ and shaken at 120 rpm. For co-expression of the MHC I[Fos] and TAPBPR[Jun], 5 mL expression medium was supplemented with 75 µg β2m-HLA-A*68:02[Fos] and 75 µg TAPBPR[Jun] DNA (equimolar ratio) and mixed with a 4-fold molar excess of linear PEI (Polysciences) diluted in 5 mL expression medium. DNA used for transfection was purified using the PureYield Plasmid Midiprep System (Promega). After incubation for 30 min, the transfection mixture was added to 50 mL of HEK293-F cells at a density of $16 \times 10^6$ cells/mL. Cells were diluted to $3 \times 10^6$ cells/mL 3 hr post transfection. Protein expression was performed in the presence of 3.5 mM valproic acid (Merck Millipore) and 10 µM kifunensine (Biomol). The culture supernatant containing the secreted proteins was harvested after five days. The *Hs*UGGT1[D1316N] construct was transfected into HEK293-F cells at a density of $1.6 \times 10^6$ cells/mL. The cell suspension was diluted to $0.8 \times 10^6$ cells/mL 3 hr post transfection and was further cultured for five days.

## Expression of UGGT1 and Sep15 in insect cells

Human UGGT1[wt]-Sep15 and Sep15 were expressed in *Spodoptera frugiperda* (*Sf*21) insect cells according to standard protocols for the Bac-to-Bac system (Thermo Fisher Scientific). *Sf*21 cells were typically grown in Sf-900 II SFM medium (Thermo Fisher Scientific) at 29 °C, and a high-titer recombinant baculovirus stock was used to infect $2 \times 10^6$ cells/mL. The cell culture medium containing secreted proteins was harvested 4 days after infection.

## Purification of MHC I[Fos]-TAPBPR[Jun] complexes

The HEK293 culture supernatant was harvested at 250×g for 10 min before adding avidin (7.5 mg/L) and adjusting to 25 mM Tris-HCl pH 8.0. Affinity chromatography was performed with Strep-Tactin Sepharose slurry (1 mL/L) (IBA Lifesciences), using 1xTris buffer (25 mM Tris-HCl pH 8.0, 100 mM NaCl) as wash buffer. The tethered complex was eluted in wash buffer supplemented with 5 mM desthiobiotin, and leucine zippers were removed by overnight digestion at 4 °C with *Tobacco Etch* Virus (TEV) protease. The complex was isolated by size-exclusion chromatography (SEC) (Superdex 200 Increase

10/300, GE Healthcare) in 1×HBS buffer (10 mM Hepes-NaOH pH 7.2, 150 mM NaCl) and concentrated by ultrafiltration (Amicon Ultra 50 kDa MWCO, Merck).

## Purification of UGGT1 proteins and immunoblotting

Culture media was harvested at 500×g for 10 min prior to dialysis (Spectra/Por 7, cellulose, 3.5 kDa MWCO, Spectrum Labs) in phosphate-buffered saline (1×PBS pH 7.6: 1.8 mM $KH_2PO_4$, 10 mM $NaH_2PO_4$, 2.7 mM KCl, 140 mM NaCl) for 3 hr at 4 °C. After dialysis overnight, the supernatant was supplemented with Ni-NTA agarose slurry (1 mL/L) for affinity chromatography and incubated at 4 °C overnight. The affinity matrix was washed in 1×PBS pH 7.6, 20 mM imidazole, 5% (v/v) glycerol and eluted in wash buffer containing 200 mM imidazole. Subsequently, the buffer was exchanged to HBS buffer (1×HBS pH 7.2, 5% (v/v) glycerol) by gel filtration (Zeba Spin Desalting column, 2 mL, Thermo Fisher). For immunoblotting, UGGT1[wt]-Sep15 was separated by SDS-PAGE (12.5%, non-reducing, Coomassie-stained) and transferred onto a nitrocellulose membrane. The membrane was blocked with 3% (w/v) dried milk in PBS-T (0.1% (v/v) Tween 20 in PBS) for 1 hr, followed by incubation with an anti-UGGT1 antibody (Abcam, Cat. No. 124879, dilution 1:3000) for 1 hr. The membrane was washed three times in PBS-T and incubated with the species-specific HRP-conjugated secondary antibody.

## Purification of Sep15

The culture supernatant was precleared by centrifugation for 15 min at 2000×g before adding 50 mM Tris-HCl pH 8.0, 1 mM $NiCl_2$, 5 mM $CaCl_2$. The resulting suspension was stirred for 30 min, followed by centrifugation at 5000×g for 30 min. The supernatant was supplemented with Ni-NTA agarose slurry (1 mL/L) and incubated overnight. The affinity matrix was washed with 1×HBS supplemented with 20 mM imidazole and finally eluted in wash buffer containing 200 mM imidazole. Buffer exchange to 1×HBS and protein concentration was conducted by ultrafiltration (Amicon Ultra 10 kDa MWCO, Merck).

## UGGT1-mediated reglucosylation

MHC I-TAPBPR complexes (3 µM) were incubated with UGGT1 (0.2–6 µM) at 21 °C in 1×HBS supplemented with 200 µM UDP-glucose (UDP-Glc) and 5 mM $CaCl_2$ (1 µM). To analyze the reglucosylation in the absence of TAPBPR, MHC I loaded with a photo-cleavable peptide (pMHC I, 1 µM) was isolated by SEC, and peptide-free MHC I (1 µM) was produced by UV illumination (365 nm, 185 mW/cm², 120 s) on ice. The mixture was incubated for 10 min at room temperature before the reglucosylation was started by adding UGGT1[wt]-Sep15 (600 nM). Recovery of MHC I-TAPBPR was conducted by UV illumination of pMHC I (1 µM) in the presence of 6 µM TAPBPR on ice. Complex formation was initiated by incubation for 10 min at room temperature. To analyze the reglucosylation of MHC I with bound low-affinity (suboptimal) peptide, peptide-receptive HLA-A*68:02 was incubated with increasing concentrations (0.1–10 µM) of the low-affinity peptide ILKCLEEPSV (predicted affinity 41 µM) before starting the reglucosylation reaction by adding UGGT (600 nM). If not otherwise specified, the reaction was stopped after 60 min by adding 25 mM of EGTA, and samples were analyzed by LC-MS. The reglucosylation reaction was monitored on the $Glc_1Man_9GlcNAc_2$/$Man_9GlcNAc_2$ intensity ratio.

## Peptide loading onto MHC I

The peptide-receptive MHC I-TAPBPR[wt] complex (10 µM) was dissociated by adding a 250-fold molar excess of the high-affinity, photo-cleavable peptide ETVSKQSJ*V (J*=3-amino-3-(2-nitro)phenyl-propionic acid) prior to incubation overnight at 4 °C. The affinity of the corresponding non-photo-cleavable peptide was predicted to be 16 nM (*Reynisson et al., 2020*). Dissociated pMHC I and TAPBPR were isolated by SEC (Superdex 200 Increase 3.2/300, GE Healthcare) in 1×HBS. ETVSKQSJ*V binding to MHC I before and after UV illumination was analyzed by LC-MS.

## Pulldown using Strep-Tactin Sepharose

MHC I[Fos]-TAPBPR[Jun] complexes (5 µM) were incubated either without or with UGGT1 (3 µM) in 1×HBS containing 5 mM $CaCl_2$ for 1 h on a rotator. To analyze the calcium dependence of the reglucosylation reaction by UGGT1, 25 mM EGTA were added to the samples. To exclude unspecific binding, UGGT[wt]-Sep15 without MHC I was incubated with Step-Tactin Sepharose. UGGT1 transiently interacting with MHC I[Fos]-TAPBPR[Jun] was pulled down by incubation with Strep-Tactin Sepharose (50%

(v/v)) for 30 min. The affinity matrix was washed twice with 1xTris buffer, and proteins were specifically eluted in 1xTris buffer with 5 mM desthiobiotin. Co-eluted proteins were analyzed by SDS-PAGE (12.5%, non-reducing, Coomassie-stained).

## Peptide loading of tethered HLA-A*68:02[Fos]-TAPBPR[Jun] complexes

Purified HLA-A*68:02[Fos]-TAPBPR[Jun] complexes (3 µM) were incubated with a 250-fold molar excess of the low-affinity peptide ILKCLEEPSV (predicted affinity 41 µM) or the high-affinity, photocleavable peptide ETVSKQSJ*V (predicted affinity 16 nM), respectively, prior to incubation for 30 min at 21 °C. Excess of non-bound peptides was removed within 120 s by rapid-spin SEC (Zeba Spin Desalting Columns, 7 kDa MWCO, Thermo Scientific). Peptides bound to isolated pMHC I (3 µM) and HLA-A*68:02[Fos]-TAPBPR[Jun] complexes were analyzed by LC-MS.

## W6/32 binding to MHC I

Peptide-loaded HLA-A*68:02 molecules (1 µM) were generated by dissociation of the Jun-Fos-cleaved HLA-A*68:02-TAPBPR[wt] complex and incubated with the W6/32 antibody (2 µM, BioLegend) for 10 min at 21 °C. To analyze W6/32 binding to peptide-free HLA-A*68:02 (1 µM), pMHC I was UV-illuminated on ice (365 nm, 185 mW/cm², 120 s). Peptide-free HLA-A*68:02 was confirmed by LC-MS (see *Figure 3—figure supplement 1*). Complex formation was initiated by adding W6/32 (2 µM) either directly or after 1 hr after UV illumination. The mixture was subsequently incubated for 10 min at 21 °C and W6/32 binding to pMHC I or MHC I was analyzed by UPLC-SEC. UPLC-SEC experiments were performed on a BioAccord System (Waters) using an ACQUITY UPLC Protein BEH SEC Column, 200 Å, 1.7 µm, 2.1 mm × 150 mm (Waters) and 1xHBS buffer. UV spectra were recorded at 280 nm with a sampling rate of 10 Hz.

## LC-MS analysis

All LC-MS analyses were conducted on a BioAccord System (Waters). Peptides were analyzed on an ACQUITY UPLC Peptide BEH $C_{18}$ Column, 130 Å, 1.7 µm, 2.1 mm × 100 mm (Waters), with a linear water/acetonitrile gradient (5–80% acetonitrile in 5.0 min) complemented with 0.1% (v/v) formic acid at 60 °C, 30 V cone voltage, 0.8 kV capillary voltage, and a desolvation temperature of 550 °C in positive mode at 50–2000 m/z. Top-down LC-MS data for glycan analysis was acquired using a cone voltage of 45 V (30 V for Sep15), 1.5 kV capillary voltage, and a desolvation temperature of 500 °C on an ACQUITY UPLC Protein BEH $C_4$ Column, 300 Å, 1.7 µm, 2.1 mm × 50 mm (Waters) at 80 °C running a linear water/acetonitrile gradient (5–45% acetonitrile in 8.0 min) supplemented with 0.1% (v/v) formic acid. Mass spectra were recorded in positive polarity at 2 Hz in full scan mode at 400–7000 m/z. Reglucosylation mixtures were spun down at 10,000×$g$ for 5 min and directly injected into the system. SEC-MS measurements were performed on an ACQUITY UPLC Protein BEH SEC Column, 200 Å, 1.7 µm, 2.1 mm × 150 mm (Waters) in 20 mM ammonium acetate. Mass spectra were recorded in positive polarity at 1 Hz in full scan mode at 400–7000 m/z with a cone voltage of 42 V, capillary voltage of 1.5 kV, and a desolvation temperature of 400 °C. Masses of peptides and glycoproteins were calculated and confirmed using the software platform UNIFY (Waters). Peptides bound to pMHC I molecules were analyzed using an ACQUITY UPLC Protein BEH $C_4$ column (Waters). pMHC I molecules were on-column dissociated. Bound peptides were analyzed at 80 °C column temperature, 30 V cone voltage, 1.5 kV capillary voltage, and a desolvation temperature of 500 °C in positive mode at 400–7000 m/z. Using the Unify small-molecule workflow, the response of peptide intensities was subsequently normalized to the signal of peptide-loaded MHC I. Combined intact mass spectra of glycoproteins were deconvoluted in UNIFY utilizing the quantitative MaxEnt1 algorithm iterating to convergence with 1.0 Da model width. Spectra with high background noise were subjected to automatic baseline correction before deconvolution. Deconvoluted spectra were centroidized based on peak height and plotted in OriginPro 2022b. UV spectra were recorded at 280 nm with a sampling rate of 10 Hz.

## Acknowledgements

We thank Dr. Ines Katharina Müller for providing the photo-cleavable peptide. We thank Dr. Simon Trowitzsch, Andrea Pott, Inga Nold, and all members of the Institute of Biochemistry for discussion and many helpful comments. This research was supported by the German Research Foundation (Reinhart

Koselleck Project TA 157/12–1 and CRC 1507 – Membrane-associated Protein Assemblies, Machineries, and Supercomplexes to R.T.) and by an ERC Advanced Grant (798121 to R.T.) of the European Research Council.

## Additional information

### Funding

| Funder | Grant reference number | Author |
|---|---|---|
| Deutsche Forschungsgemeinschaft | TA 157/12-1 | Robert Tampé |
| Deutsche Forschungsgemeinschaft | CRC 1507/P18 | Robert Tampé |
| European Research Council | 798121 | Robert Tampé |

The funders had no role in study design, data collection and interpretation, or the decision to submit the work for publication.

### Author contributions

Lina Sagert, Conceptualization, Data curation, Formal analysis, Validation, Investigation, Visualization, Methodology, Writing - original draft, Writing - review and editing; Christian Winter, Data curation, Formal analysis, Validation, Investigation, Visualization, Methodology; Ina Ruppert, Maximilian Zehetmaier, Data curation; Christoph Thomas, Conceptualization, Data curation, Formal analysis, Supervision, Validation, Investigation, Visualization, Writing - original draft, Project administration, Writing - review and editing; Robert Tampé, Conceptualization, Data curation, Formal analysis, Supervision, Funding acquisition, Validation, Investigation, Visualization, Writing - original draft, Project administration, Writing - review and editing

### Author ORCIDs

Ina Ruppert http://orcid.org/0000-0002-0448-2815
Christoph Thomas http://orcid.org/0000-0001-7441-1089
Robert Tampé http://orcid.org/0000-0002-0403-2160

### Decision letter and Author response

Decision letter https://doi.org/10.7554/eLife.85432.sa1
Author response https://doi.org/10.7554/eLife.85432.sa2

## Additional files

### Supplementary files
- MDAR checklist
- Source data 1. This file contains all orignal gels, immunoblots, and data.

### Data availability

All data generated or analyzed during this study are included in the manuscript and Supporting Source Data file deposited at: https://doi.org/10.5281/zenodo.7505421.

The following dataset was generated:

| Author(s) | Year | Dataset title | Dataset URL | Database and Identifier |
|---|---|---|---|---|
| Sagert L, Winter C, Ruppert I, Zehetmaier M, Thomas C, Tampé R | 2023 | The ER folding sensor UGGT1 acts on TAPBPR-chaperoned peptide-free MHC I | https://doi.org/10.5281/zenodo.7505421 | Zenodo, 10.5281/zenodo.7505421 |

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
