## [Editor Report]

This valuable study reports a complete in-vitro system where different steps and direct interactions between different components of MHC I maturation can be monitored, hence leading to a better mechanistic understanding of MHC I maturation. The evidence supporting the findings is solid and the methods, data and analyses broadly support the claims with only minor weaknesses. This work will be of interest to immunologists and biochemists.

---

## [Decision Letter]

**Decision letter after peer review:**

Thank you for submitting your article "The ER folding sensor UGGT1 acts on TAPBPR-chaperoned peptide-free MHC I" for consideration by *eLife*. Your article has been reviewed by 2 peer reviewers, and the evaluation has been overseen by a Reviewing Editor and Tadatsugu Taniguchi as the Senior Editor. The following individual involved in the review of your submission has agreed to reveal their identity: Scheherazade Sadegh-Nasseri (Reviewer #2).

Essential revisions:

1) Please consider comment 2 of Reviewer 1: " This may be a phenomenon specific to HLA-A*68:02 as previous studies have shown that HLA-B*08 associated with a low-affinity peptide is recognized by UGGT1 in the absence of TAPBPR (Zhang et al., 2011). It would be good if the authors carried out a similar assay using another allele, in particular B*08, to see if this is something specific to their assay or to the allele."

2) Please consider comment 4 of Reviewer 1. "Since the expression of mutant TAPBPR is known to impact the peptides presented by HLA-A*68:02, it is imperative that the authors analyze whether any peptides are bound to the complexes purified from the cells."

*Reviewer #1 (Recommendations for the authors):*

Although this in vitro study breaks down the quality control of MHC-I by reglucosylation to its principal components, there are some issues that need to be addressed, which may also further contribute to the understanding of this process.

1. It is assumed that the pMHC-I/TAPBPR complex isolated is 'empty', or without associated peptide. O'Rourke et al. and Sagert et al. both show that these complexes can be loaded with high-affinity peptides but they do not actually carry out MS analysis to show conclusively that no peptide is associated with the purified complexes. If low-affinity peptides are indeed associated, that could influence the interpretation of the data.

2. It is striking in Figure 3 that peptide-free MHC-I molecules are not recognized by UGGT1 in the absence of TAPBPR. This may be a phenomenon specific to HLA-A*68:02 as previous studies have shown that HLA-B*08 associated with a low-affinity peptide is recognized by UGGT1 in the absence of TAPBPR (Zhang et al., 2011). It would be good if the authors carried out a similar assay using another allele, in particular B*08, to see if this is something specific to their assay or to the allele. One concern is how long after peptide removal by UV irradiation is reglucosylation attempted, which ideally would be addressed in a kinetic experiment, adding back TAPBR at different times. The enzyme reaction itself takes one hour. However, after UV irradiation and peptide cleavage empty class I molecules are likely unstable and the β2m component, even though it is covalently linked to the heavy chain, may locally 'dissociate', leaving a denatured substrate that is no longer susceptible to the action of UGGT1. Asking whether the MHC-I molecules react post peptide cleavage with conformationally susceptible antibodies, such as w6/32, or heavy chain reactive antibodies, such as HC10, could potentially address this question. UGGT1 is known to be incapable of glucosylating substrates that are heavily denatured. The presence of TAPBPR may stabilize a susceptible conformation.

3. 'Empty' MHC-I molecules vs. MHC-I associated with low-affinity peptides may be recognized differently by UGGT1, perhaps because these peptides also sufficiently stabilize a susceptible population. Can the addition of a low-affinity peptide after photo cleavage restore susceptibility to UGGT1 activity in the absence of TAPBPR? Points 2 and 3 together address the issue of whether TAPBPR and low-affinity peptides are functionally equivalent, i.e. do both bind to MHC-I-β2m dimers to stabilize a UGGT1-susceptible conformation?

4. It is surprising that the MHC-I associated with TAPBPRC97A is glucosylated to a greater extent by UGGT1 (Figure 4B). Since the expression of mutant TAPBPR is known to impact the peptides presented by HLA-A*68:02, it is imperative that the authors analyze whether any peptides are bound to the complexes purified from the cells. If HLA-A*68:02/TAPBPRC97A complexes are associated with low-affinity peptides, (as suggested by Neerincx et al., 2017), it may explain why there is greater glucosylation of the MHC-I heavy chain seen in Figure 4B.

*Reviewer #2 (Recommendations for the authors):*

The authors do a good job of providing the background information in the introduction section of the manuscript; however, they miss out on spelling out the outstanding question that led to the work presented here. As written, the manuscript reads as an exercise done to check out the previous findings. Also, the discussion is mainly a reiteration of the Results sections. It would be better to discuss their results in comparison with the literature focusing on differences and similarities and how the new findings can be useful to move the field forward.

---

## [Author Response]

Essential revisions:1) Please consider comment 2 of Reviewer 1: " This may be a phenomenon specific to HLA-A*68:02 as previous studies have shown that HLA-B*08 associated with a low-affinity peptide is recognized by UGGT1 in the absence of TAPBPR (Zhang et al., 2011). It would be good if the authors carried out a similar assay using another allele, in particular B*08, to see if this is something specific to their assay or to the allele."2) Please consider comment 4 of Reviewer 1. "Since the expression of mutant TAPBPR is known to impact the peptides presented by HLA-A*68:02, it is imperative that the authors analyze whether any peptides are bound to the complexes purified from the cells."

We thank the reviewers for their positive feedback and valuable input, which helped us to further refine our manuscript. Below you will find our point-by-point response to the reviewers' comments.

Reviewer #1 (Recommendations for the authors):Although this in vitro study breaks down the quality control of MHC-I by reglucosylation to its principal components, there are some issues that need to be addressed, which may also further contribute to the understanding of this process.1. It is assumed that the pMHC-I/TAPBPR complex isolated is 'empty', or without associated peptide. O'Rourke et al. and Sagert et al. both show that these complexes can be loaded with high-affinity peptides but they do not actually carry out MS analysis to show conclusively that no peptide is associated with the purified complexes. If low-affinity peptides are indeed associated, that could influence the interpretation of the data.

We thank the reviewer for pointing out this aspect which is rarely addressed in literature. We expressed and purified tethered HLA-A*68:02^Fos^-TAPBPR^Jun^ complexes, thus keeping the peptide exchange catalyst TAPBPR in direct proximity to MHC I. We hypothesize that neither low- nor high affinity peptides can bind to this tethered complex, unless the Jun-Fos zipper is cleaved. In our assays, the Jun-Fos tether is removed in a peptide-free environment. To confirm that peptides cannot stably associate with tethered HLA-A*68:02^Fos^-TAPBPR^Jun^ we incubated these MHC I-chaperone complexes with high concentrations (750 µM, 250-fold molar excess) of low-affinity peptide (ILKCLEEPSV, *K*_d_ ≈ 41 µM) or high-affinity peptide (ETVSKQSJ*V, *K*_d_ ≈ 10 nM). Subsequently, the bound peptide was analyzed by intact protein LC-MS analysis (new Figure 4—figure supplement 1). These data demonstrated that peptide did not bind neither to the HLA-A*68:02^Fos^-TAPBPR^wt-Jun^ nor to the HLAA*68:02^Fos^-TAPBPR^C97A-Jun^ complex. We therefore conclude that acer purification no peptides are prebound in the complex, which could have a major effect on data interpretation.

2. It is striking in Figure 3 that peptide-free MHC-I molecules are not recognized by UGGT1 in the absence of TAPBPR. This may be a phenomenon specific to HLA-A*68:02 as previous studies have shown that HLA-B*08 associated with a low-affinity peptide is recognized by UGGT1 in the absence of TAPBPR (Zhang et al., 2011). It would be good if the authors carried out a similar assay using another allele, in particular B*08, to see if this is something specific to their assay or to the allele. One concern is how long after peptide removal by UV irradiation is reglucosylation attempted, which ideally would be addressed in a kinetic experiment, adding back TAPBR at different times. The enzyme reaction itself takes one hour. However, after UV irradiation and peptide cleavage empty class I molecules are likely unstable and the β2m component, even though it is covalently linked to the heavy chain, may locally 'dissociate', leaving a denatured substrate that is no longer susceptible to the action of UGGT1. Asking whether the MHC-I molecules react post peptide cleavage with conformationally susceptible antibodies, such as w6/32, or heavy chain reactive antibodies, such as HC10, could potentially address this question. UGGT1 is known to be incapable of glucosylating substrates that are heavily denatured. The presence of TAPBPR may stabilize a susceptible conformation.

We thank the reviewer for the helpful suggestion to analyze the β2m-MHC I-hc stability with the conformation-specific antibody W6/32. Our data demonstrate that W6/32 binds to pMHC I and peptide-free MHC I after photocleavage of peptide and release. We also examined the long-term stability of peptide-free MHC I by addition of W6/32 one hour after UV-illumination and were still able to detect a complete shift of the peptide-receptive MHC I population. We conclude that the peptide-receptive MHC I molecules remain stable for the duration of the reglucosylation experiment. It is important to mention that the stability of peptide-receptive MHC I molecules has already been addressed in our TAPBPR rescue experiment. Here, the formation of an MHC I-TAPBPR complex restored the reglucosylation by UGGT1. The new data on the stability of peptide-receptive MHC I were appended to Figure 3—figure supplement 2.

We also consider to repeat the assay with a structurally and mechanistically well-characterized H2D^b^-TAPBPR complex. However, due to the glycan complexity (three glycan sites) we could not assign mass peaks to specific mono-glucosylated glycans. Regarding HLA-B*08:01, it was recently shown that this allomorph displays no detectable interaction with TAPBPR (Ilca et al., 2019; Sun et al., 2023). Thus, it seems likely that some allomorphs can be read out by UGGT1 in a TAPBPR-independent manner. We included this aspect in the discussion.

Moreover, the expression procedure, the complex assembly, and the photo-triggered approach leading to peptide-free HLA-B*08:01 would have to be re-established for this allomorph, which is not feasible in the revision time. In addition, there is the likelihood that a peptide-free HLA-B*08:01 with a defined Man_9_GlcNAc_2_ glycan cannot be isolated via the HLA-B*08:01-TAPBPR complex. It is worth mentioning that our studies were performed with a defined glycan due to glycosylation engineering in human cells using kifunensine. An almost complete reglucosylation of the MHC I allomorph was shown in complex with the chaperone TAPBPR. In previous studies, HLA-B*08:01 was expressed in insect cells, and the glycan modifications of this allomorph were not analyzed, and the reglucosylation was not stoichiometrically quantified with regard to the glycan (Zhang *et al.* 2011). Thus, it is difficult to compare TAPBPR-dependent and -independent allomorphs, different glycan modification, and reglucosylation assays in both studies. We have covered this aspect in the discussion of our revised manuscript.

3. 'Empty' MHC-I molecules vs. MHC-I associated with low-affinity peptides may be recognized differently by UGGT1, perhaps because these peptides also sufficiently stabilize a susceptible population. Can the addition of a low-affinity peptide after photo cleavage restore susceptibility to UGGT1 activity in the absence of TAPBPR? Points 2 and 3 together address the issue of whether TAPBPR and low-affinity peptides are functionally equivalent, i.e. do both bind to MHC-I-β2m dimers to stabilize a UGGT1-susceptible conformation?

We would like to thank the reviewer for the smart idea to demonstrate the crucial function of TAPBPR in UGGT1-mediated reglucosylation by additional experiments. To address the concerns, we incubated the isolated, UV-exposed (and peptide-free) HLA-A*68:02 with increasing concentrations of low-affinity peptide (0.1-10 µM of ILKCLEEPSV). However, the addition of a low-affinity peptide did not restore UGGT1 ac2vity towards HLA-A*68:02, and we could not demonstrate a concentration-dependent effect. These results further emphasize the indispensable role of TAPBPR in UGGT1mediated reglucosylation. We added these data to the manuscript (new figure 3—figure supplement 3).

4. It is surprising that the MHC-I associated with TAPBPRC97A is glucosylated to a greater extent by UGGT1 (Figure 4B). Since the expression of mutant TAPBPR is known to impact the peptides presented by HLA-A*68:02, it is imperative that the authors analyze whether any peptides are bound to the complexes purified from the cells. If HLA-A*68:02/TAPBPRC97A complexes are associated with low-affinity peptides, (as suggested by Neerincx et al., 2017), it may explain why there is greater glucosylation of the MHC-I heavy chain seen in Figure 4B.

As explained in point 1, we analyzed the peptide-binding competence of both HLA-A*68:02^Jun^TAPBPR^Fos^ complexes (wt and C97A) (see new figure 4—figure supplement 1). Since both complexes did not bind peptides, we conclude that the observed differences in UGGT1-mediated reglucosylation were not caused by the peptide-loading status of MHC I.

Reviewer #2 (Recommendations for the authors):The authors do a good job of providing the background information in the introduction section of the manuscript; however, they miss out on spelling out the outstanding question that led to the work presented here. As written, the manuscript reads as an exercise done to check out the previous findings. Also, the discussion is mainly a reiteration of the Results sections. It would be better to discuss their results in comparison with the literature focusing on differences and similarities and how the new findings can be useful to move the field forward.

We have modified the discussion, and we now focus on the comparison with the latest findings and consider future aspects in the quality control of antigen presentation. In the revised manuscript we compared our data on a TAPBPR-dependent allomorph with an MHC I allele that has recently been shown not to interact with TAPBPR and is therefore likely to be TAPBPR-independently reglucosylated. Using glycosylation engineering and producing peptide-free MHC I molecules and MHC I-TAPBPR complexes with a structurally defined Man_9_GlcNAc_2_ glycan, we can decipher the minimal requirements of an almost complete reglucosylation by UGG1. Our established TAPBPR-mediated reglucosylation assay of a TAPBPR-dependent allomorph is furthermore compared with the literature using a TAPBPR-independent allomorph.